# Incidence of the Brownian Relaxation Process on the Magnetic Properties of Ferrofluids

**DOI:** 10.3390/nano14070634

**Published:** 2024-04-05

**Authors:** Lili Vajtai, Norbert Marcel Nemes, Maria del Puerto Morales, Kolos Molnár, Balázs Gábor Pinke, Ferenc Simon

**Affiliations:** 1Department of Physics, Institute of Physics, HUN-REN-BME Condensed Matter Research Group, Budapest University of Technology and Economics, Műegyetem rkp. 3., H-1111 Budapest, Hungary; vajtai.lili@edu.bme.hu (L.V.); simon.ferenc@ttk.bme.hu (F.S.); 2Departamento de Física de Materiales, Universidad Complutense de Madrid, 28040 Madrid, Spain; 3Department of Nanoscience and Nanotechnology, Instituto de Ciencia de Materiales de Madrid (ICMM-CSIC), 28049 Madrid, Spain; puerto@icmm.csic.es; 4Department of Polymer Engineering, Faculty of Mechanical Engineering, Budapest University of Technology and Economics, Műegyetem rkp. 3., H-1111 Budapest, Hungary; molnar@pt.bme.hu (K.M.); pinke@pt.bme.hu (B.G.P.); 5HUN–REN–BME Research Group for Composite Science and Technology, Műegyetem rkp. 3., H-1111 Budapest, Hungary; 6MTA-BME Lendület Sustainable Polymers Research Group, Műegyetem rkp. 3., H-1111 Budapest, Hungary; 7Institute for Solid State Physics and Optics, HUN-REN Wigner Research Centre for Physics, P.O. Box 49, H-1525 Budapest, Hungary

**Keywords:** ferrofluid, Brownian relaxation, Néel relaxation, magnetometry, hyperthermia, nanomagnetism

## Abstract

Ferrofluids containing magnetic nanoparticles represent a special class of magnetic materials due to the added freedom of particle tumbling in the fluids. We studied this process, known as Brownian relaxation, and its effect on the magnetic properties of ferrofluids with controlled magnetite nanoparticle sizes. For small nanoparticles (below 10 nm diameter), the Néel process is expected to dominate the magnetic response, whereas for larger particles, Brownian relaxation becomes important. Temperature- and magnetic-field-dependent magnetization studies, differential scanning calorimetry, and AC susceptibility measurements were carried out for 6 and 13.5 nm diameter magnetite nanoparticles suspended in water. We identify clear fingerprints of Brownian relaxation for the sample of large-diameter nanoparticles as both magnetic and thermal hysteresis develop at the water freezing temperature, whereas the samples of small-diameter nanoparticles remain hysteresis-free down to the magnetic blocking temperature. This is supported by the temperature-dependent AC susceptibility measurements: above 273 K, the data show a low-frequency Debye peak, which is characteristic of Brownian relaxation. This peak vanishes below 273 K.

## 1. Introduction

Nanomagnetic ferrofluids remain a focus of interest due to several unexpected phenomena as well as some intriguing applications such as lubricants [1], medical imaging contrast materials [2], or magnetothermal heating substances in medical therapy [3,4,5]. Although the fundamental magnetic properties of ferrofluids have been studied [6,7,8,9], their high-frequency magnetic response is less explored [10,11]. Knowledge of these is crucial for magnetic hyperthermia applications, as it sets the optimal working conditions (e.g., frequency) for radiofrequency irradiation to avoid overlap with unwanted side effects (e.g., heating of the body caused by eddy currents) [12].

The relaxation time approximation is a widely used picture for the description of the time-dependent magnetic response of ferrofluids. It predicts that two different relaxation mechanisms are present in these materials, which determine the time scale of the response. These are Brownian relaxation, which is related to mechanical rotations, and Néel relaxation, which is connected to magnetic domain rotations. In the approximation, these are believed to act independently and therefore can be substituted with an effective relaxation process, characterized by the time scales of both underlying mechanisms. These quantities are heavily dependent on particle size, and the properties of the effective value result in that (except for a small size range) either one or the other characteristic time scales (and the corresponding processes) dominate the effective behavior. This model has its limitations: both the distribution of particle size and interactions between particles can result in deviation from the proposed behavior, and the expressions usually used for describing characteristic times are only valid in small magnetic fields. Despite all these limitations, the relaxation time approximation seems to describe dilute ferrofluids with well-defined particle sizes quite accurately [1,13,14].

Our work focuses on the prevalence of the Brownian relaxation process, as this phenomenon is less well described in the literature, whose focus is on dry nanoparticle systems, where the process is hindered as it is related to the mechanical rotation of the particles. In some ferrofluids, however, this appears to be the dominant process. In addition, by freezing the carrier liquid, one can basically “switch off” the Brownian process, allowing for a convenient investigation of the phenomenon. Based on our results, Brownian relaxation is present in the liquid phase and vanishes upon freezing the fluid.

Néel relaxation is different as it originates from the magnetic anisotropy energy and is present in both liquid and solid phases (above the so-called blocking temperature, which can be thought of as a temperature threshold, above which the thermal energy overcomes the magnetic energy barrier between preferred directions) [15]. The time scale of this mechanism depends exponentially on both the inverse temperature and the particle size, resulting in drastically varying relaxation time values [13,14,16].

The frequency-dependent power absorption in nanomagnetic fluids is governed by the imaginary part of the dynamic susceptibility (or $\chi^{\prime \prime }$). In its simplest mono-exponential form, resulting in a Debye relaxation behavior, it is characterized by a single relaxation time, τ. This, in turn, is governed by the two different aforementioned relaxation processes. In the first one, the particle is static and its magnetization rotates (known as the Néel process), and in the second one, the entire particle rotates (known as the Brownian process) while its magnetization is unchanged with respect to the particle [14]. The first one is typical for smaller particles (typically with diameters below 10 nm, this value is dependent both on the atomic structure and the geometry of the nanoparticles), and the second one occurs for larger particles [1,13,17,18].

Temperature- and magnetic-field-dependent studies in nanomagnetic ferrofluids have focused mostly on Néel-dominated samples [5,9,16,19,20,21,22,23,24] using DC magnetometry, with a limited effort for larger particles with Brownian relaxation [8,9,21]. These effects were also theoretically studied for various models, including ferrohydrodynamics [1,5,14,25,26], the Fokker–Planck equation [27], or the egg model using the Landau–Lifsitz–Gilbert theory [21,28,29]. The most common approach in the investigation of dilute ferrofluids is to use linear response theory [21], but it is also possible to consider higher-order terms in the magnetic response [29]. Several studies have focused on the preparation and characterization of a particular nanomagnetic ferrofluid using magnetometry [4,18,21,30,31,32]. AC susceptibility was also studied using both frequency- and time-domain approaches [10,11,27,33,34,35], but no information exists on the temperature- and frequency-dependent contributions of the two processes, although it is highly relevant for clinical applications in nanomagnetic hyperthermia [13,36,37].

We investigated the properties of ferrofluids containing magnetite (Fe_3_O_4_) nanoparticles of four different sizes (6 nm, 8 nm, 10.6 nm, 13.5 nm); however, we only discuss the results of the two extrema, a 6 nm and a 13.5 nm sample, in detail (for further results and the comparison of the other nanoparticles, see Appendix A). These samples demonstrate the largest differences in behavior and are the most typical for the Brownian and the Néel relaxation processes, as the characteristic time scales of both mechanisms depend on the particle size in a system of nanoparticles.

We performed temperature-dependent DC magnetometry, including field-cooled (FC) and zero-field-cooled (ZFC) measurements, and identified the main differences in the two sets of results caused by different relaxation mechanisms being dominant. The experiments were accompanied by differential scanning calorimetry (DSC) measurements to precisely monitor the freezing of water and to correlate its effect with the magnetometry results. The temperature-dependent AC susceptibility measurements clearly identify the contributions of the two types of processes. The presence of a characteristic, temperature-dependent magnetic hysteresis (taken at low magnetic fields) around the solvent freezing temperature is identified as a straightforward fingerprint for Brownian relaxation.

The main aim of this article is to identify and explain the effect of the Néel and Brownian relaxation processes on the magnetic response of ferrofluids. Our findings make it possible to provide a size estimate of the nanoparticles based solely on the magnetic response, without the need for microscopy.

## 2. Materials and Methods

The magnetic nanoparticles were obtained following the co-precipitation protocol described by Massart [38] by introducing small modifications to control particle size. The particle size was measured with transmission electron microscopy (TEM),with a JEOL JEM 1011 transmission electron microscope (Peabody, MA, USA) with Gatan ES1000Ww camera (Pleasanton, CA, USA), resulting in 5.8 ± 1.5 nm for the 6 nm sample and 13.2 ± 4 nm for the 13.5 nm sample. Most reported data were taken on sufficiently diluted samples with 3 mg/mL concentration, although we studied the concentration dependence up to 30 mg/mL, where some influence was detected, and the results are discussed in the Appendix A. Further investigations of similar nanoparticles were conducted in previous studies [8,39,40].

As stated in the Introduction, the main assumption of our study is the validity of the relaxation time approximation in the investigation of our samples. This approach is reliable for sufficiently diluted ferrofluids with well-defined particle sizes. The first assumption is validated by the fact that, upon measuring more dense fluids (see Appendix A), there is no qualitative difference in the results. The particles are also assumed to be superparamagnetic [41] in the aqueous, liquid environment, which is supported by the hysteresis loops, which exhibit zero area.

A Quantum Design MPMS3 device (San Diego, CA, USA) was used in DC scan mode for the static magnetometry measurements. Typically, 10–20 μL of the investigated ferrofluids were permanently sealed with a torch in quartz tubes. The magnetic field offset due to the history-dependent trapped superconducting flux was corrected by comparing it to the hysteresis cycles of a paramagnetic Pd reference.

To study the so-called magnetic blocking effect, we performed zero-field-cooled (ZFC) and field-cooled (FC) experiments; for the latter, we used 10 kOe. We employ SI units throughout the manuscript, except for the magnetometer output, which is in CGS units. The temperature ramp rate was set to 2 K/min, and the calorimetric studies validated that this protocol results in the proper thermalization of the solutions.

Differential scanning calorimetry (DSC) measurements were performed with a TA Instruments DSC Q2000 device (New Castle, DE, USA). Sample holders with unfixed lids fabricated for this instrument were filled with 10–20 μL of the investigated ferrofluid suspensions. During measurements, the samples were uniformly cooled and heated between 25 °C and −50 °C, and the required heat flow was recorded. Multiple measurement cycles were performed with both the heating and the cooling rate at 1, 2, 5, 10, and 20 K/min. Below 5 K/min (speed of both cooling and heating), results were no longer affected by further slowing down the rate. This implies that the 2 K/min ramping rate in the magnetometry also results in proper thermalization. Peaks in the heat flow data are signs of first-order phase transitions, such as the freezing or the melting of the carrier fluid.

A small amount of carrier fluid may evaporate during lengthy measurements (a typical duration being 30 min), but repeated cycling revealed that it did not affect the DSC results: neither the heat flow nor the observed transition temperatures (see Appendix A).

Frequency-dependent magnetic susceptibility was measured using the AC measurement system (ACMS) option of the Quantum Design PPMS system (San Diego, CA, USA) between 10 Hz and 10 kHz using various excitation field amplitudes (0.1–10 Oe) in a zero DC magnetic field. The AC susceptibility was not affected by the excitation field amplitude.

## 3. Results and Discussion

We primarily discuss nanoparticles with nominal diameters of d=6 nm and d=13.5 nm, as these show the largest contrast between the Néel and Brownian relaxation behaviors. We also discuss intermediate-diameter samples in the Appendix A. Figure 1 shows the magnetization curves for the two types of samples at room temperature (magnetization is normalized by the mass of iron contained in the sample). Saturation occurs in a relatively low magnetic field (a few 100 Oe), which indicates superparamagnetic behavior, with the number of correlated spins being around 100–1000. The magnetic saturation of superparamagnetic nanoparticles is expected to follow a Langevin function [32].

The Langevin function is the classical limit of the Brillouin function with spin quantum number *S* in the limit S→∞ [42]. In our case, the ensembles of ferromagnetically coupled individual spins can be considered “superspins”, thus approaching the classical limit.

The assumption of non-interacting particles was verified by performing measurements on more dense fluids (see Appendix A). Moreover, as can be validated by some simple calculations, the nanoparticles contain a few thousand spins (which stem from the magnetization of the magnetite), so approximating the particles as infinitely large spins seems to be well established.

The analytic expression for the Langevin function is the following [42]:(1)$$L\left(x\right)=\coth\left(x\right)-\frac{1}{x},$$
which describes the sample magnetization as $M=M_{s}L\left(x\right)$, where $M_{s}$ is the saturated magnetization and $x=\frac{H\mu_{\mathrm{part}}}{k_{B}T}$. Here, $k_{B}$ is the Boltzmann constant, T=300K is the temperature, and $\mu_{\mathrm{part}}$ is the magnetic moment of the superparamagnetic particle.

Figure 1 shows good Langevin fits for the measured magnetization curves, yielding reasonable saturation magnetization values (95.8 emu/g Fe for the 6 nm sample and 73.6 emu/g Fe for the 13.5 nm sample). The values in the literature for $M_{s}$ are between 80 emu/g Fe_3_O_4_ [43] and 86 emu/g Fe_3_O_4_ [44], which can be converted to 111 emu/g Fe and 119 emu/g Fe. The deviation of the measured data from these nominal values may stem from some uncertainty in the measurement of the nanoparticle amount in the suspensions and also from a slightly altered oxidation state of Fe ions on the nanoparticle surfaces. The latter is known to reduce the magnetization when freshly prepared magnetite transforms into maghemite with time [45].

One expects that $\mu_{\mathrm{part}}$ is about 8–10 times higher for the 13.5 nm sample than it is for the 6 nm sample (based on the volume ratio of the two particles), which is clearly not the case in the Langevin fits, as quoted in Figure 1. The Langevin fits do not account for any variation in nanoparticle diameter but are more sensitive to the value of $M_{s}$, and the result can be refined by inspecting the low-field data. In fact, we observe a deviation between the data and the fits in Figure 1 (probably due to particle size distribution in the samples), which becomes more apparent when the numerical derivative of the data and the fits are inspected (the data are shown in the Appendix A). This also allows us to obtain $\mu_{\mathrm{part}}$ from the low-field derivative values. Given that the Langevin function can be approximated as
(2)$$L\left(x\right)=\frac{x}{3}-\frac{x^{3}}{45}+𝒪\left(x^{5}\right)$$
near the origin, the numerical derivative of the data at zero field also gives an estimate for $\mu_{\mathrm{part}}$ as
(3)$${\left.\frac{dM}{dH}\right|}_{H=0}=\frac{M_{s}\mu_{\mathrm{part}}}{3k_{B}T}$$

Using the numerical derivative values of 0.14 emu/g Oe (6 nm sample) and 0.68 emu/g Oe (13.5 nm sample), we obtain $\mu_{\mathrm{part}}=1.82$ × $10^{-23}$ J/Oe (6 nm sample) and $\mu_{\mathrm{part}}=1.14$ × $10^{-22}$ J/Oe (13.5 nm sample). Clearly, both values are larger than their counterparts from the Langevin fits, the change being almost a factor of 8 for the 13.5 nm sample. This difference between the permeability based on the saturation magnetization and the low-field slope is probably related to the distribution of particle size (see Appendix A). Alternatively, aggregation of the small grains may also influence the result. This affects the saturation and the permeability differently: saturation magnetization is independent of the particle size, whereas the low-field permeability depends linearly on the number of correlated spins in a superparamagnetic particle.

These refined $\mu_{\mathrm{part}}$ values allow us to determine the number of correlated spins and the size of the nanoparticles, given that the magnetic moment per Fe_3_O_4_ unit is 4.1 $\mu_{B}$ in magnetite [46]. We obtain d= 9 ± 3 nm =9(3) nm (6 nm sample) and d=12±5 nm =12(5) nm (13.5 nm sample) and the number of correlated spins N=4.8 × $10^{3}$ (6 nm sample) and N=3 × $10^{4}$ (13.5 nm sample). These diameter values match reasonably well with those determined using TEM (which gave d= 5.8 ± 1.5 nm for the smaller particles and d= 13.2 ± 4 nm for the larger ones, see Appendix A; however it is important to note that in the case of the results from TEM, the uncertainty refers to the standard deviation parameter of the Gaussian curve fitted to the particle size histogram, whereas in the case of the calculation, it denotes the uncertainty of the mean particle size estimation), especially since the magnetic data-based analysis may be affected by several factors, including inaccuracies in the concentration and mass measurements and a non-magnetic or a reduced-magnetic shell on the surfaces of the nanoparticles [45].

Figure 2 shows the magnetic hysteresis curves at various temperatures for both samples. For the 6 nm sample, the coercive field is zero at room temperature, with hysteresis appearing only below 100 K. This means that this sample remains superparamagnetic down to 100 K, as expected for small-sized nanoparticles. In contrast, the 13.5 nm sample displays a hysteresis below 273 K, with the coercive field becoming progressively larger upon cooling. This observation shows that the freezing of water also hinders the nanoparticle tumbling, i.e., it results in the freezing out of Brownian relaxation.

The above observations and the drastically different behavior of the two samples are further supported by a series of field-cooled (FC) and zero-field-cooled (ZFC) experiments, which are shown in Figure 3. In both cases, the measurements were performed upon heating at 100 Oe after cooling either in zero field or at 10 kOe. The curves are offset manually (in both cases by not more than 5 emu/g Fe) so that the data match at 300 K. This is necessary as the amount of trapped flux is unknown following the field cooling, as mentioned in Section 2. We cannot exclude either that, in the liquid form, the ferrofluid moves in the sealed capillary, which also affects the measurements.

In the ZFC experiment, the two samples show markedly different behavior: there is a large jump in the magnetization (about 50%) for the 13.5 nm sample around 273 K, but the magnetization in the 6 nm sample shows a negligible change at this temperature. This observation is fully compatible with the expected dominance of Brownian relaxation for the 13.5 nm sample [47]. The magnitude of the jump is comparable in the FC experiment for both samples, an effect whose explanation is less evident: one would expect that the 6 nm sample would not be affected at all by the water freezing. The observation may be due to a differing preferred direction for the small nanoparticles in 10 kOe.

For the 6 nm sample, a maximum in the ZFC data magnetization is observed at around 50 K, which is usually associated with the so-called blocking temperature [48], $T_{B}$. This corresponds to the blocking of the rotation of sample magnetization, i.e., below this point, the Néel process is also hindered and the sample no longer shows superparamagnetic behavior. For the 13.5 nm sample, the blocking temperature is expected to be above room temperature [49]. In fact, extrapolation of the low-temperature ZFC for the 13.5 nm sample (dashed line in Figure 3) shows a merger with the FC data, i.e., $T_{B}$ is above room temperature. Clearly, the blocking temperature cannot be observed in the 13.5 nm sample above 300 K as nanoparticle tumbling prevails. However, this blocking temperature would be observed for the 13.5 nm sample if the nanoparticles were embedded in a solid matrix.

In addition to the conventional FC and ZFC measurements, we found that thermal hysteresis measurements in moderate magnetic fields (100 Oe) also provide valuable information into the magnetization behavior. In contrast to the high magnetic field in the FC studies (10 kOe), the applied 100 Oe is insufficient to fully orient the nanoparticle magnetization and it is rather a “readout” magnetic field. Therefore, it allows us to read out the magnetic state of the nanoparticles without strongly affecting it. The data are shown in Figure 4: in the measurement, we first cooled and then warmed the sample in the same unchanged 100 Oe magnetic field.

The figure shows shaded areas for the freezing and melting of the same sample suspensions using differential scanning calorimetry, DSC. The details of the DSC analysis are provided in the Appendix A. A freezing–melting hysteresis is expected given the first-order nature of this phase transition. The DSC method delivers range rather than exact temperatures for the onset of these processes. As a cautionary note, we mention that melting/freezing in the vicinity of the nanoparticles may not necessarily occur at the same temperature as melting/freezing in the bulk of the solvent [8,21]; however, in our measurement, the two effects occur at approximately the same temperatures.

We observe a thermal hysteresis for both types of samples between the freezing and melting temperatures. However, it is much larger for the 13.5 nm sample, as expected when Brownian relaxation dominates the sample magnetization. The magnetization in the 13.5 nm sample is higher in the liquid state, which means that Brownian relaxation allows for the rotation of the nanoparticle along a preferred direction of the nanoparticle magnetization with respect to the external field.

The data also show an interesting, albeit yet-unexplained, behavior for the 13.5 nm sample. One would expect the magnetization to increase as the solvent melts but to stay constant when the solvent freezes since the particles could rotate to better align their magnetic moments with the applied field in the liquid. In contrast to this expectation, the magnetization decreases suddenly below the freezing of water. We surmise that the Brownian process allows for increased magnetization due to the dynamical alignment of the particles.

We also studied the concentration dependence of this effect (the data are shown in the Appendix A) to find out whether the particle–particle interactions may give rise to this, but we did not observe such an effect. We discuss below that the vanishing of part of the sample magnetization upon the freezing of the solvent for the 13.5 nm sample corroborates the AC susceptibility measurements. Our observations also indicate that the low-temperature ground state of the material when cooled at 100 Oe is insensitive to the thermal history.

An excellent benchmark method to reveal Brownian relaxation is the type of data shown in Figure 4: it only requires the application of a moderate magnetic field, whose magnitude does not need to be precisely calibrated.

The thermal hysteresis studies indicate that there exists a temperature range (between about 250 and 280 K) where both the liquid and solid states can be studied at the same temperature, depending on the thermal history. To this end, we studied magnetic hysteresis at 270 K, where both phases are realized upon cooling or warming, and the result is shown in Figure 5. No magnetic hysteresis is observed for the 6 nm sample in either case, implying that the magnetization direction is unaffected by the state of the surroundings for small-sized nanoparticles. This is significantly different for the 13.5 nm sample, where a pronounced hysteresis appears when the solvent is frozen, which is in full agreement with the dominance of the Brownian process for this particle size.

Figure 6 shows the frequency-dependent real and imaginary parts of the AC magnetic susceptibility at various temperatures for both samples. The figure also contains illustrations of the dominant relaxation processes of the particles. In Figure 6b, the Néel process is demonstrated, which means that the magnetic response is the result of the magnetization rotating with respect to the particles, while the particles themselves do not exhibit mechanical motion. This mechanism is not affected by the freezing of the carrier liquid. The Néel process is dominant in the magnetic response of smaller particles. Figure 6d illustrates Brownian relaxation, which is related to the mechanical rotation of the particles upon subjecting the material to a magnetic field, while the magnetization remains immobile with respect to the particles. As freezing the carrier liquid immobilizes the particles, in a frozen ferrofluid, this mechanism is not expected to be present; therefore, the only remaining phenomenon is the Néel process (as both mechanisms are present in all ferrofluids but with different time scales, and the shorter one is dominant). This is the case for larger nanoparticles.

The complex AC magnetic susceptibility spectra are expected to follow a Lorentzian curve [13,35,36,37] from a mono-exponential Debye model as
(4)$$\tilde{\chi }\left(\omega \right)=\chi^{\prime }\left(\omega \right)+i\chi^{\prime \prime }\left(\omega \right)=\chi_{0}\left[\frac{1}{1+\omega^{2}\tau^{2}}+i\frac{\omega \tau }{1+\omega^{2}\tau^{2}}\right]$$
here, $\chi_{0}$ is the static susceptibility, the angular frequency is ω=2πf, and the relaxation time, τ, is a combination of the Néel ($\tau_{Né\mathrm{el}}$) and Brownian ($\tau_{\mathrm{Brown}}$) relaxation times:(5)$$\frac{1}{\tau }=\frac{1}{\tau_{Né\mathrm{el}}}+\frac{1}{\tau_{\mathrm{Brown}}}$$

The linear response theorem dictates that the real and imaginary parts are Hilbert transforms of one another.

For the smaller-diameter nanoparticles (6 nm sample), freezing the solvent (300 K vs. 270 K) does not change the magnetic susceptibility appreciably: $\chi^{\prime }$ slightly increases, which is consistent with the steepening of the hysteresis curves in Figure 2 in the small magnetic field regime. There is no observable sign of Debye characteristics in this frequency range; the characteristic frequency of Néel relaxation processes (i.e., $1/\tau_{Né\mathrm{el}}$) is typically much larger [13,36,37].

For the larger nanoparticles (13.5 nm sample) in the liquid phase, there is a visible Debye function corresponding to Brownian relaxation with a relaxation time $\tau_{\mathrm{Brown}}$. Upon freezing, this contribution vanishes, and the curves become similar in shape and magnitude to the ones recorded for the 6 nm sample. This implies the presence of a residual Néel process even when the sample is frozen.

A global Lorentzian fit for data taken at 300 K for the 13.5 nm sample was performed, and the result is shown in Figure 6 along with the fit parameters. The fit is not accurate, probably due to a distribution in the particle sizes (see Appendix A). Clearly, in our frequency range, we cannot determine the frequency of the Néel process. Our fit gave a relaxation time of $\tau_{\mathrm{Brown}}=12.6$μs. The seminal paper of Ortega and Pankhurst [13] tabulates that such a Brownian relaxation time corresponds to a particle size of about 12 nm based on relaxation time calculations performed for similar nanoparticle systems to the ones investigated here. This result is in good agreement with our nominal particle size of 13.5 nm. We also note that Brownian relaxation is determined by the hydrodynamic diameter, which may be different from the apparent nanoparticle diameter (see Appendix A).

The vanishing of Brownian relaxation and its sensitivity to temperature have a direct consequence for the application of nanoparticles in, e.g., nanomagnetic hyperthermia. Therefore, knowledge of frequency-dependent magnetic susceptibility is crucial to determining optimal irradiation conditions. In turn, this information may provide a spectroscopic fingerprint on the nature of the surrounding tissue: when the nanoparticles are immobile, the AC magnetization frequency spectrum is significantly altered.

## 4. Conclusions

We observed drastic diameter-dependent differences in the magnetometric and calorimetric responses of well-characterized ferrofluidic magnetite nanoparticle water suspensions. Namely, we identified the fingerprints of both magnetic and thermal hysteresis on the nanoparticle magnetism when the Néel or Brownian relaxation processes dominate. These observations may serve as benchmarks for sample characterization solely based on the magnetization data. We identified that a low magnetic field measurement through the solvent freezing/melting temperature provides the simplest means to characterize whether Brownian relaxation dominates the magnetic properties. We also identified the fingerprints of the different relaxation processes in the frequency-dependent magnetization data, which has direct therapeutic relevance for medical applications.

Our main findings are that we observed and recognized the signature of both the Néel and Brownian relaxation processes on the magnetic response of ferrofluids and discussed them based on the properties of these mechanisms. We also provided a size estimation of the nanoparticles contained in a ferrofluid sample based on low-magnetic-field magnetometry measurements at room temperature without the need for microscopy.

These findings enable the prediction of further characteristics of the magnetic response of a ferrofluid. The applications of ferrofluids strongly rely on the dominant relaxation process, thus our method may be suitable for investigating the applicability of other ferrofluids. It is important to note that the validity of these predictions is tied to the type of solvent, the type of nanoparticle, and its morphology. The dominant relaxation mechanism is expected to display a different size dependence in such cases.

The properties of ferrofluids are still far from being fully understood, so there are still plenty of unanswered questions regarding their properties. One possible direction of our future study is, for instance, to investigate other magnetic susceptibility measurement techniques.

## Figures and Tables

**Figure 1 nanomaterials-14-00634-f001:**
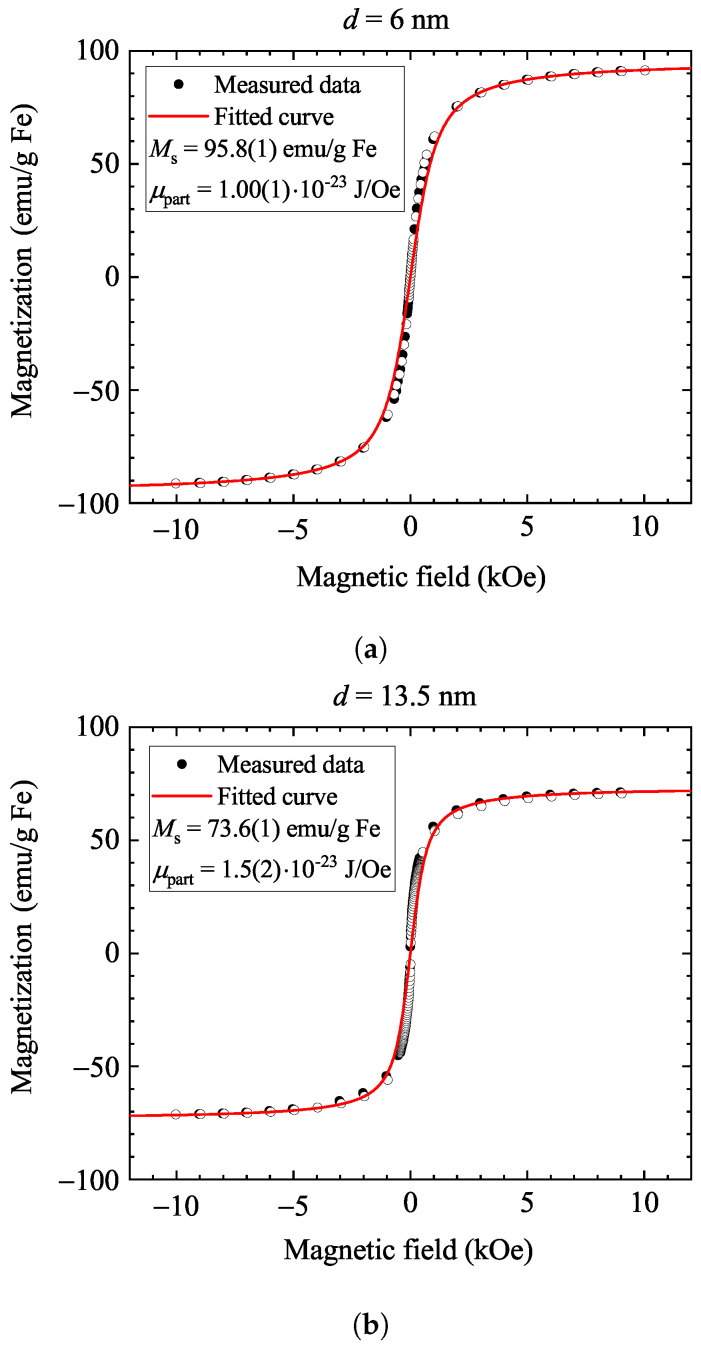
Magnetization curves for the two types of magnetite samples recorded at 300 K: containing 6 nm particles (**a**) and 13.5 nm particles (**b**). The fitted Langevin functions are shown with solid curves and fitting parameters. Full circles denote data recorded at decreasing magnetic field, whereas data with open symbols were taken at increasing magnetic field.

**Figure 2 nanomaterials-14-00634-f002:**
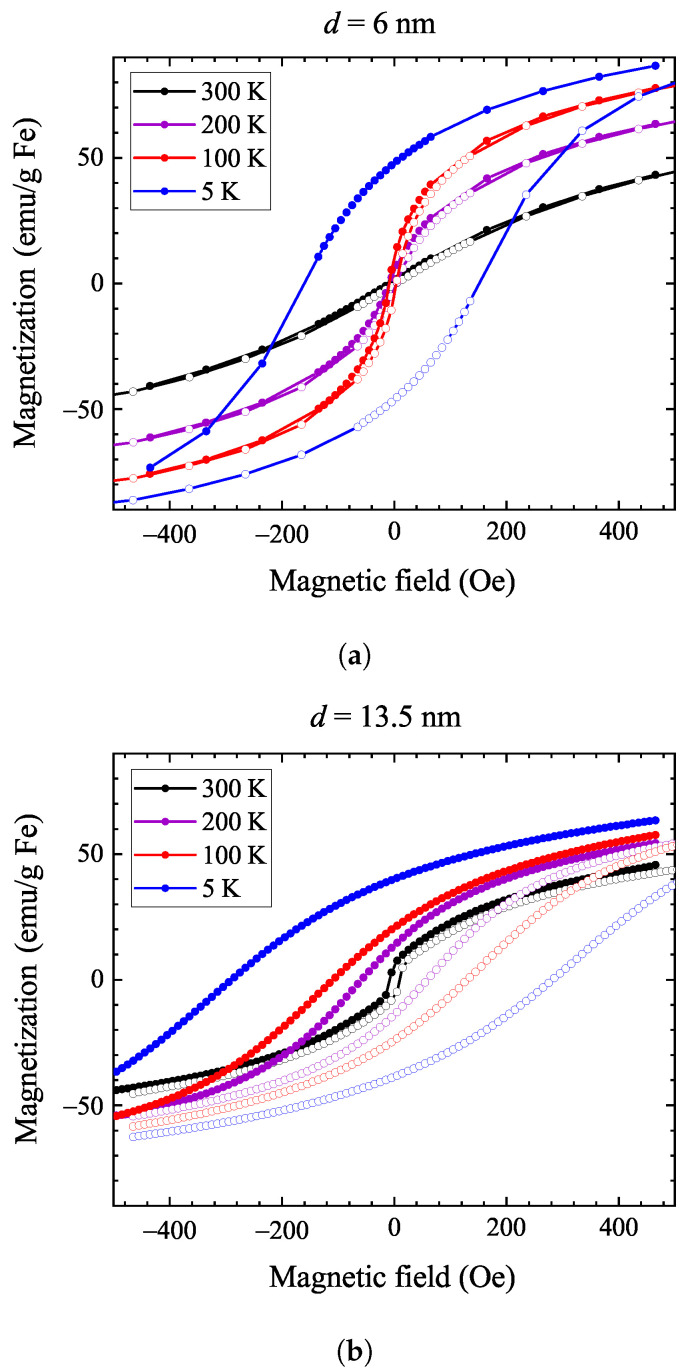
Magnetization hysteresis loops at different temperatures for the 6 nm sample (**a**) and the 13.5 nm sample (**b**). Note that hysteresis develops only below 100 K for the 6 nm sample, whereas it appears immediately below 300 K for the 13.5 nm sample. Full symbols denote data recorded at decreasing magnetic field, while data with open symbols were measured at increasing magnetic field.

**Figure 3 nanomaterials-14-00634-f003:**
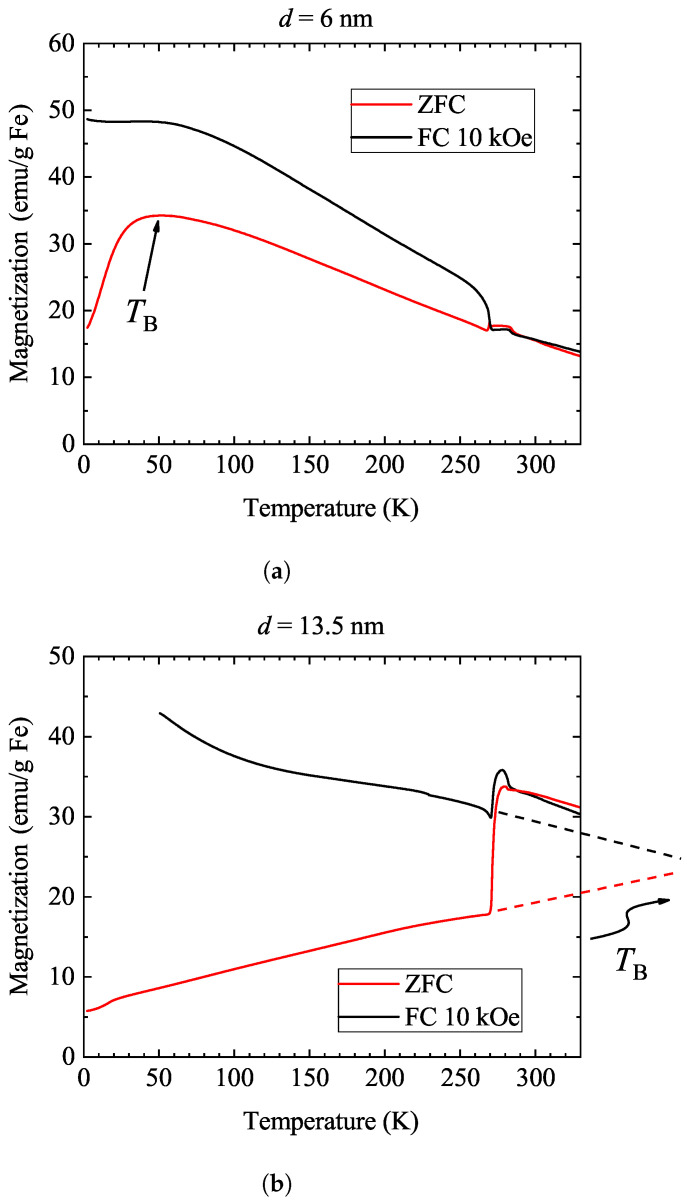
Zero-field-cooled and field-cooled (at 10 kOe) magnetization measurements for the 6 nm (**a**) and 13.5 nm (**b**) samples. Both measurements were performed upon warming at 100 Oe. The approximate position of $T_{B}$ is indicated for the 6 nm sample; however, it lies above 300 K for the 13.5 nm sample. For the latter sample, an extrapolation to the low-temperature data is shown with dashed lines, and a higher-than-room temperature, $T_{B}$, is indicated.

**Figure 4 nanomaterials-14-00634-f004:**
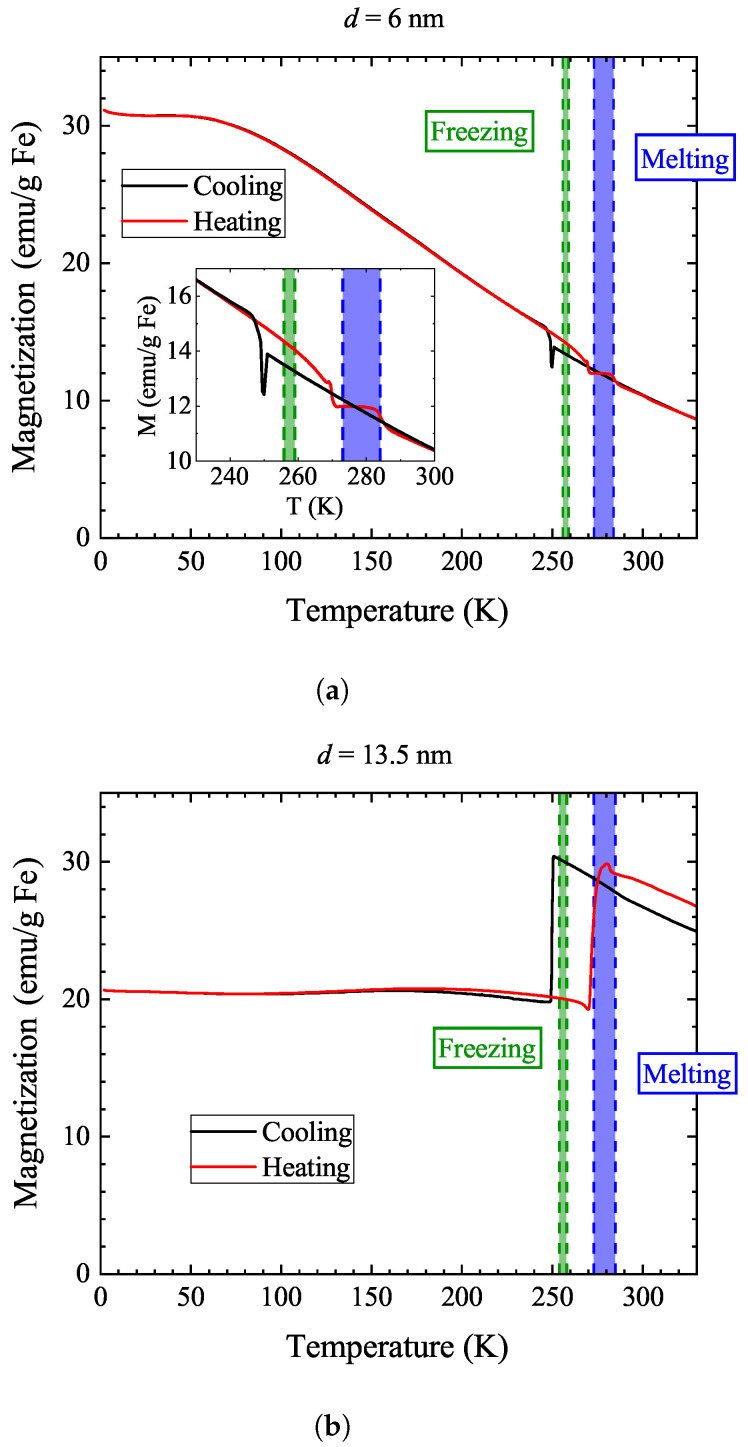
Thermal hysteresis studied by monitoring the magnetization in a moderate magnetic field (100 Oe) as a function of temperature in cooling and warming for the 6 nm sample (**a**) and the 13.5 nm sample (**b**). The temperature ranges where freezing and melting occur are obtained from the DSC studies and are indicated by shaded areas. The inset of (**a**) shows the phase transition temperature ranges of the 6 nm sample in more detail.

**Figure 5 nanomaterials-14-00634-f005:**
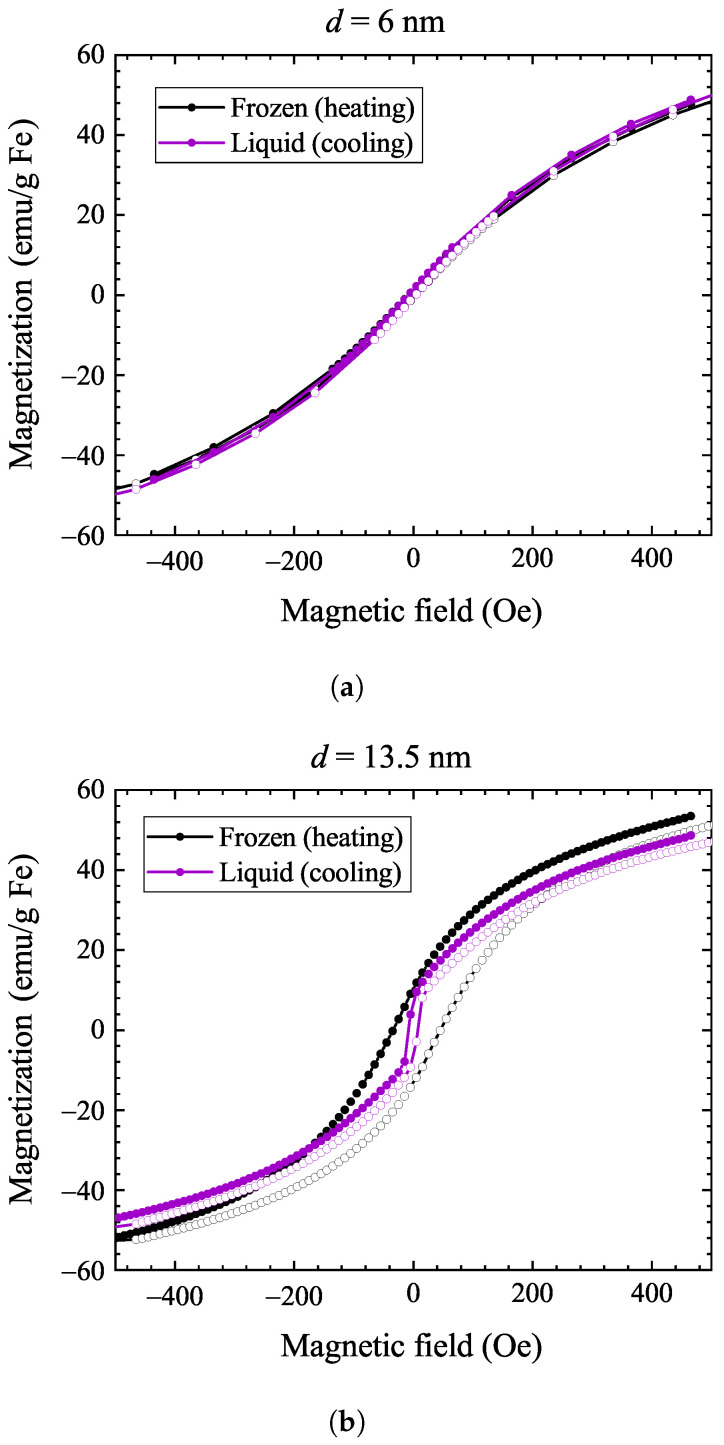
Magnetic hysteresis curves at 270 K in liquid and solid states in the case of the 6 nm sample (**a**) and the 13.5 nm sample (**b**). Liquid state: purple curves, frozen state: black curves. Data denoted by full symbols were recorded while decreasing the magnetic field, whereas open symbols stand for data measured at increasing magnetic field.

**Figure 6 nanomaterials-14-00634-f006:**
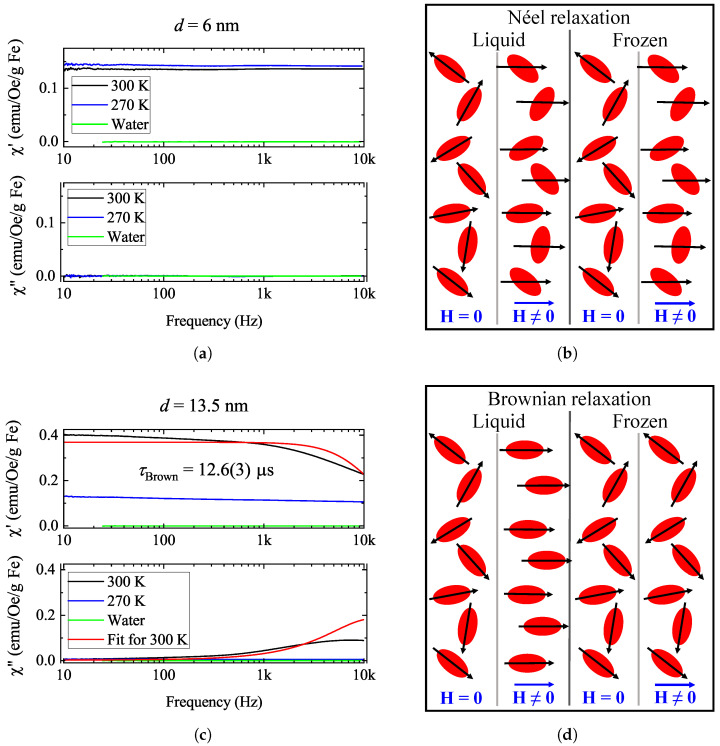
Frequency spectra of the real and imaginary parts of AC magnetic susceptibility for a few temperatures across the water freezing/melting for the 6 nm (**a**) and 13.5 nm (**c**) samples and the illustration of the dominant relaxation mechanisms (Néel relaxation for the smaller particles (**b**) and Brownian for the larger ones (**d**)). For the 13.5 nm sample, a Lorentzian fit is provided from a mono-exponential Debye model. In the illustrations of the relaxation processes, nanoparticles are denoted by red shapes (they are elongated only to make it easier to demonstrate their orientation), the magnetic field by a blue arrow, and the magnetization of the particle by a black arrow.

## Data Availability

Dataset available on request from the authors.

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
