# Peer review of "Incidence of the Brownian Relaxation Process on the Magnetic Properties of Ferrofluids"

_nanomaterials, 2024, doi:10.3390/nano14070634_

Round 1

Reviewer 1 Report

Comments and Suggestions for Authors

(1) The “3. Results” subtitle is suggested to be changed to “3. Result and Discussion”.

(2) Advise in each plot, d=6 nm is indicated by (a) and d = 13.5 nm by (B), the (top) change to (a) and the (bottom) change to (b).

(3) The nanoparticles average diameters were 6, 8, 10.6, and 13.5 nm in the manuscript, but why the performance of only d=6 nm and d=13.5 nm are provided.

Reviewer 2 Report

Comments and Suggestions for Authors

This work deal with Incidence of the Brownian relaxation process on the magnetic properties of ferrofluids and is very interesting for relate-field researchers. For next process, author should response to questions and improve the manuscript.

1. Introduction section, I recommend that authors would show originality compared to other researches such as magnetic particles.

2. I can find only magnetic properties data of samples including FE-SEM. Authors should showed other measurement results such as XRD and TEM and explain the effect of particle size and temperature in the samples.

3. In ferrofluids, is there only Brownian relaxation? How about Neel relaxation?

Comments on the Quality of English Language

I recommend Authors would check typos and missing in your manuscript. 

Reviewer 3 Report

Comments and Suggestions for Authors

This work investigates the effects of the Néel relaxation vs. Brownian relaxation on water based ferrofluids containing Fe3O4 magnetic nanoparticles. Mainly, the particle sizes of 6 and 13.5 nm were studied since relaxation is dominated by the Néel (Brownian) process on the 6nm (13.5nm) size. Temperature-dependent DC magnetometry and AC susceptibility measurements are performed, and magnetic and thermal hysteresis curves are obtained. Fingerprints of the Brownian relaxation are found for the 13.5 nm sample. In addition, frequency dependent magnetization data are obtained and discussed.

The significance of this work is that a low magnetic field measurement around freezing and melting temperature allows characterizing if the magnetic property of the system is due to the Brownian process.

As far as I read the manuscript, the methodology used in this work sounds logical and reasonable. The manuscript and supplementary materials provide sufficient data. 

I have a few minor comments for the authors.

One minor suggestion is to add a diagram of the Néel regime and the Brownian regime -- something like Fig. 1 (c) in Ota and Takemura, 

J. Phys. Chem. C 2019, 123, 47, 28859–28866.

In the abstract, the authors write the study was done for nanoparticles with 6, 8, 10.6, and 13.5 nm. However, the main manuscript only shows 6 and 13.5 nm cases. If you are mentioning the 8 and 10.6 nm nanoparticles in the abstract, you are expected to show and discuss these results in the main manuscript. So you should remove it from the abstract. 

On page 4, the authors discuss the fitting with the Langevin function and approximate Langevin function. I'm curious how much the curve fitting errors are because that would tell you the accuracy of the curve fitting.

In line 139, it says "d=8.8 nm (6-nm sample) and d=16 nm (13.5-nm sample)," but I did not understand what this means. If I read this correctly, these are nanoparticle sizes determined using mu_part from the fitting(?), and those values differ from the size determined from the TEM measurement(?). Why would they be so different?

Figure 4 caption has an incomplete sentence. "The inset of the" 

Reviewer 4 Report

Comments and Suggestions for Authors

In the current manuscript, the authors attempted to study the water-based ferrofluids containing magnetite (Fe3O4) nanoparticles of well-controlled and distinctively different diameters. The nominal average diameters were 6, 8, 10.6, and 13.5 nm. The Néel relaxation dominates the 6-nm sample, whereas the Brownian relaxation is expected to be the dominant process in the 13.5-nm sample. The authors performed temperature-dependent DC magnetometry, including field-cooled (FC) and zero-field-cooled (ZFC) measurements, and identified the main differences in the two sets of results caused by different relaxation mechanisms being present. The experiments were accompanied by DSC (differential scanning calorimetry) measurements to precisely monitor the freezing of water and to correlate its effect with the magnetometry results. Temperature-dependent AC susceptibility measurements clearly identified the contributions of the two types of processes. The presence of a characteristic, temperature-dependent magnetic hysteresis (taken at low magnetic fields) around the solvent freezing temperature was identified as a straightforward fingerprint for Brownian relaxation.

The paper is well structured and has certain merits for publication. However, prior to any final decision, the authors should suitably address the following concerns:

- The authors should clearly express the assumptions and limitations of their models and methods.

- Do you have any evidence to prove that the evaporation of carrier fluid did not affect the DSC results?

- What is the superiority of the proposed model compared to the other ones in the literature?

- Please describe the “Langevin function” and its credibility to predict the magnetic saturation of nanoparticles.

- The new contribution of the current study should be further clarified and highlighted.

- Can you explain why within the freezing process the Debye function of larger nanoparticles vanishes? More discussion on this statement is needed for clarification.

- The literature review should be enriched, for instance, by discussing recent references regarding the properties of ferrofluids. The following articles in the literature are suggested to be included and discussed.

-  https://doi.org/10.22190/FUME211228019H

- DOI: 10.22055/JACM.2021.36722.2892

- A subtitle cannot start with a figure. Actually, as a general rule, a figure should not appear in a paper before it has been referred to in the text.

- More discussions on the results are needed to present the major outcomes of the present study.

- The authors use the form such as “16.5-nm sample”. I see no reason for the dash there.

- Conclusions should not only repeat what has been done in the work. What can be generally concluded based on your work? You need to address this question. Also, are there any limitations of the work you have done? This may be addressed as well. And you need to provide some clear directions for the future work in this field. 

Round 2

Reviewer 2 Report

Comments and Suggestions for Authors

Dear Authors

Authors responded to all question and comments.

I have one more question. Figures (b) and (d) show schematic diagram for dominant relaxation mechanisms and is it right?

Comments on the Quality of English Language

Please check typo and mistake in your manuscript. 

Reviewer 4 Report

Comments and Suggestions for Authors

The authors have suitably revised the manuscript. It is recommended for publishing as it is. 

Author Response

We are thankful to the Reviewer for the positive evaluation, and all the insightful remarks and suggestions they have made.